# The Reduction in the Deformation of HDPE Composites Using Self-Lubricating Fillers in an Aqueous Environment

**DOI:** 10.3390/polym14030433

**Published:** 2022-01-21

**Authors:** Chuanbo Liu, Shutian Liu, Conglin Dong, Chengqing Yuan, Xiuqin Bai

**Affiliations:** 1School of Mechanical and Electronic Engineering, Wuhan University of Technology, Wuhan 430070, China; Lchb72@whut.edu.cn; 2Polymer Technology, Department of Mechanical Engineering, Eindhoven University of Technology, P.O. Box 513, 5600 MB Eindhoven, The Netherlands; 3National Engineering Research Center for Water Transportation Safety, Reliability Engineering Institute, Wuhan University of Technology, Wuhan 430063, China; ycq@whut.edu.cn (C.Y.); xqbai@whut.edu.cn (X.B.); 4School of Transportation and Logistics Engineering, Wuhan University of Technology, Wuhan 430063, China

**Keywords:** HDPE composite, Si_3_N_4_, surface deformation, fluctuation reduction

## Abstract

Reducing the deformation of polymer matrix materials can decrease the fluctuation of coefficient of friction (COF), and friction-induced vibration and its amplitudes. HDPE composites with T-ZnOw as a fixed strengthening filler were modified with the addition of Si_3_N_4_ particles at different concentrations. The COFs, wear rates, micro-morphologies, and friction-induced vibrations were obtained by conducting sliding tests against carbon steel balls in an aqueous environment at a low velocity and high load. The mechanism of the reduction in frictional fluctuation due to the addition of Si_3_N_4_ particles was revealed through the frictional responses. The results demonstrated that 4 wt% addition of Si_3_N_4_ in HDPE can enhance the strain–stress property and improve the lubrication by forming a lubricating film. Therefore, the surface deformation and the fluctuations of COFs and its vibrations were reduced. The aggregation phenomenon and reduced strain–stress response at a high concentration of Si_3_N_4_ disrupted the positive fluctuating reduction, and resulted in a rough surface with severe tearing and cracking deformations. Additionally, it led to fluctuating wear behaviors with high COF and vibrations. The results obtained in this study can elucidate the effects of adding Si_3_N_4_ particles to enhance lubrication in polymer composites. Additionally, the results provide a new research method for designing and manufacturing polymer-based composites with low friction-induced fluctuations.

## 1. Introduction

Polymer matrix composites are used in various fields, such as automotive, aircraft, aerospace, and marine machinery, due to their exceptional mechanical and physical properties [1,2,3,4]. Polymeric composites have replaced conventional metal parts in ships because they are light-weight and wear-resistant. Additionally, polymer-based water-lubricated stern bearings exhibit excellent performance [5,6,7,8]. However, factors, such as unstable loads, high temperature, and boundary lubrication during the operation of ship stern bearings, cause the surface deformation of polymer-based materials. Therefore, this results in a severe friction process, which affects the reliability and stability of ships [9,10,11,12].

Moreover, polymer-based water-lubricated bearings are prone to surface deformations at a low velocity and heavy load conditions, which causes vibrations and fluctuations during operation [13,14,15,16]. Several methods were used to reduce the surface deformation of polymers. Fillers with lubricating effects can be utilized as lubricating enhancers to improve the lubricating conditions on frictional interfaces. Molybdenum disulfide (MoS_2_), black phosphorus (BP), and graphene have a unique two-dimensional structure, and they have been widely used to modify polymers by reducing friction-induced deformations to enhance lubrication [17,18,19,20]. Although fillers with two-dimensional structures can significantly increase the lubricating condition of polymer-based composites, chronic friction and poor working conditions might rupture the positive lubricating environment and cause severe friction damage.

Therefore, single two-dimensional particles modifying fillers used for the reduction in surface deformation were replaced with three-dimensional particles with unique structures, and flower-like particles were adopted to fulfill such demands [21,22,23,24]. Tetra-needle-like ZnO whiskers (T-ZnOw) can reduce the surface deformation of high-density polyethylene (HDPE) composites due to its unique three-dimensional structure which can restore worn surfaces [25,26]. In addition, the utilization of flower-like fillers, such as MoS_2_, can transform an applied external force into internal friction within the polymers by increasing the contact area, which decreases the surface deformation tendency [27,28,29,30]. Furthermore, a few particles with chemical potential have been studied to improve the lubricating effects of polymers in particular lubricating conditions. For instance, it was demonstrated that silicon nitride (Si_3_N_4_) can enhance the surface hardness and lubricating effects of polymers in an aqueous environment [31,32].

Although the lubricating fillers and flower-like fillers have been used to modify materials to reduce surface deformation individually, the synergistic effect of these two different fillers on improving the tribological property has rarely been conducted. Therefore, in this study, the T-ZnOw particles, which proved to perform excellent deforming resistance on the polymer surface, were adopted as enhancers with static content on the HDPE composites. Additionally, the Si_3_N_4_ particle was selected as a lubricating enhancer due to its stable physical properties and exceptional chemical reacting potential in the aqueous environment [33,34]. The results obtained in this study can explore the possibility of Si_3_N_4_ as a new lubricating filler in an aqueous environment, and reveal the synergistic effect of two different fillers on improving the tribological property in the polymeric material.

## 2. Methods and Experiments

### 2.1. Experimental Materials

HDPE particles with an average diameter of 2 mm were selected as the matrix material due to their good thermoplastic properties. These particles are shown in Figure 1B. Tetra-needle-like ZnO filler (size: 20–25μm) was used as the enforcement filler due to its unique three-dimensional structure (Figure 1A). Si_3_N_4_ particles with a diameter of 5 μm were selected as the self-lubricating particle to modify the HDPE composite (Figure 1C). The particle fillers were procured from the Aladdin Co., Ltd. (Shanghai, China). T-ZnOw, Si_3_N_4,_ and HDPE particles were mixed using a double screw extrusion granulation machine (ZC-65D, Nanjing Zhicheng Rubber Machinery Co., Ltd., Nanjing, China) to obtain a uniform mixture. The mixture was melted and formed into cuboid-shaped modified HDPE specimens in a lapping tool using a DKM injection machine (DQ-180T, DeQun Machine Co., Ltd., Guangzhou, China), as shown in Figure 1D. The friction and mechanical tests were conducted on cylindrical- and dumbbell-shaped samples shown in Figure 1F. The energy-dispersive spectroscopy (EDS) curve and the element distribution maps are shown in Figure 1E. It can be observed that silicon (Si) and zinc (Zn) were present in the mixture and were uniformly distributed in the HDPE composite.

The HDPE composite with 6 wt% mass content of T-ZnOw was used in [26], and it exhibited a low and stable coefficient of friction (COF) response, which demonstrates a reduction in the surface deformation. Therefore, the content of T-ZnOw selected in this study was 6 wt% and it acted as an enhancer. Additionally, the filler mass content of Si_3_N_4_ was variable, and its different contents were set to 0 wt%, 2 wt%, 4 wt%, 6 wt%, and 8 wt%, respectively.

### 2.2. Experimental Apparatus and Sliding Wear Tests

The friction experiments were conducted using a commercial ball-on-pin friction testing machine, as shown in Figure 2A (R-tec Tribo-meter, Rtec instruments Inc., San Jose, CA, USA). A 10 mm diameter carbon steel ball, which is an extensively used engineering frictional metal material, was selected as the frictional pair (Figure 2C). The HDPE specimen had a cylindrical shape with a diameter of 30 mm and a height of 8 mm (Figure 2B). During the tribological experiment, the carbon steel ball was maintained in a stationary position at the top, while the lower HDPE specimens were rotated at a velocity of 0.021 m/s (Figure 2D). The vertical load was 80 N and the radius of rotation was set to 10 mm. Additionally, a vibration sensor was fixed on the upper fixture to measure the vibration induced in the HDPE-based composites during the wear and friction process (also shown in Figure 2D). The aqueous environment was simulated by continuously adding distilled water between the frictional interfaces during the experiments to ensure sufficient water availability. The experiments were conducted for 1800 s and the test data were recorded at an interval of 0.001 s. A new polymer plate and carbon steel ball were used for each experiment. In addition, all the tests in this study were repeated thrice to ensure repeatability.

### 2.3. Measurement Techniques and Procedures

The surface topography was examined using confocal laser scanning microscopy (CLSM) (VX-X1000, KEYENCE, Osaka, Japan), and the surface element analysis of the cuboid specimens was performed using energy-dispersive spectroscopy (EDS) (X-act one, Oxford, UK). Additionally, the micro-morphology of the wear scratch was observed using a scanning electron microscope (SEM) (VEGA3, TESCAN, Brno, Czech Republic). The vibration experimental result was tested and analyzed using the Bruel & Kjaer testing machine (Sensor 4535-B, B & K Ltd., Nærum, Denmark). Furthermore, the surface hydrophilicity of the polymer with distilled water was characterized by contact angle measurement (DSA 100, KRUSS GmbH, Hamburg, Germany) at 23 °C. The surface roughness value was obtained using a white light interferometer (Micro Xam, ADEP Hase Shift, Inc., Tucson, AZ, USA). The mechanical properties, such as the shore hardness and strain–stress response, were tested using a universal testing machine (Instron 1341, Instron Ltd., Boston, MA, USA).

## 3. Experimental Results

### 3.1. Properties of the Modified Composites

The contact angle, shore hardness, and strain–stress responses of the modified HDPE composites are shown in Figure 3. It can be observed from Figure 3A that the addition of Si_3_N_4_ did not enhance the water-absorbing property of the HDPE composites, because the contact angles were approximately 90°, which meant hydrophilicity and hydrophobicity [35,36]. The shore hardness of the HDPE composite increased with the addition of Si_3_N_4_. The shore hardness without the addition of Si_3_N_4_ was 79 A, whereas the shore hardness increased to 86 A with 8 wt% Si_3_N_4_ addition, which resulted in a hardened surface (Figure 3B). In addition, the yield stress and strain hardening properties (Figure 3D) can be obtained from the complete strain–stress curves of the modified HDPE composites (Figure 3C). Initially, the yield stress of the pure composite increased from 50 MPa to a maximum value of 53 MPa, with the addition of 4 wt% Si_3_N_4_. The yield stress decreased to 48 MPa and 46 MPa with the addition of 6 wt% and 8 wt% Si_3_N_4,_ respectively. Similarly, the strain hardening initially presented an increasing trend to a peak of 3.9 MPa with the addition of 4 wt% Si_3_N_4_, and then decreased with the addition of the filler. These results demonstrate that the addition of Si_3_N_4_ above a particular concentration severely affects the strain–stress properties, which might be attributed to the agglomeration of fillers.

### 3.2. The Frictional Coefficient Results of the Modified Composites

Figure 4 depicts the COF responses and its amplitudes between the HDPE plates and the carbon steel ball at 0.021 m/s and 80 N in a water-lubricated environment. The COF curves of the pure HDPE/T-ZnOw composite (Figure 4A) had an average value of approximately 0.086 during the testing period, and the COF amplitudes (ΔA) were in the range of 0.08–0.09. The addition of 2 wt% Si_3_N_4_ reduced and maintained a COF of approximately 0.079 with reduced fluctuations in the range of 0.077–0.081 (Figure 4B). Moreover, after a short down-trend in the initial wear, a 4 wt% addition of Si_3_N_4_ in the HDPE composite decreased the COF to a stable value of 0.065, and significantly narrowed the COF amplitudes with fluctuations of 0.002 (Figure 4C). However, when the content of Si_3_N_4_ was increased to 6 wt%, the COF curve of the HDPE composite exhibited unstable and increasing trends (Figure 4D), in which a maximum value of 0.11 was observed in combination with a sharp increase in COF amplitudes (ΔA) in the range of 0.082–0.103. Furthermore, COF amplitudes in the range of 0.12–0.09 were observed for the HDPE composite with the addition of 8 wt% Si_3_N_4_, and the maximum value of COF reached 0.122 (Figure 4E). The average COFs and COF amplitudes of the modified HDPE composites are depicted in Figure 4F. The average COF initially decreased from 0.085 for the pure HDPE composite, to a minimum value 0.062 with the addition of 4 wt% Si_3_N_4_, and then it sharply increased when the Si_3_N_4_ content was equal to or exceeded the T-ZnOw content. The maximum value was 0.11, with the addition of 8 wt% Si_3_N_4_. Similarly, the COF amplitudes (ΔA) initially declined to the lowest value (fluctuating from 0.002 to 0.003) with the addition of 4 wt% Si_3_N_4_, before it rose to the highest value (fluctuating from 0.1 to 0.12) with the addition of 8 wt% Si_3_N_4_. The COF responses obtained indicated that the addition of Si_3_N_4_ can decrease the COF and COF amplitude, which performed excellently with a 4 wt% Si_3_N_4_ addition. However, a sustained growth of Si_3_N_4_ contents broke this positive phenomenon and resulted in the increase in COF with fluctuating amplitudes.

### 3.3. The Friction-Induced Vibration of the Modified Composites

The friction-induced vibration response of the modified HDPE composites at 0.021 m/s and 80 N in an aqueous environment is shown in Figure 5. HDPE with only T-ZnOw exhibited regular small vibration amplitudes in the range of −0.25–0.31 m/s^2^, during the 10-s testing period (Figure 5A). Furthermore, the vibration amplitudes dropped to a smaller scope in the range of −0.26–0.23 m/s^2^ with the addition of 2 wt% Si_3_N_4_ to the HDPE composite (Figure 5B). The addition of 4 wt% Si_3_N_4_ to the HDPE resulted in the smoothest vibration response with small amplitudes in the range of −0.16–0.13 m/s^2^, and it exhibited an excellent vibrational behavior (Figure 5C). However, when the Si_3_N_4_ content was equal to or greater than that of the T-ZnOw content (6 wt%), the vibrational behavior was severe and exacerbated. Two fluctuating points were observed at 1502 s (in the range of −0.75–0.51 m/s^2^) and at 1508 s (in the range of −0.78–0.77 m/s^2^) for the HDPE composite when the Si_3_N_4_ content was 6 wt% (Figure 5D). Moreover, the addition of 8 wt% Si_3_N_4_ led to significant amplitudes on the vibration response in the range of −0.52–0.76 m/s^2^ and −1.3–1.2 m/s^2^ between 1503–1504 s and 1506–1508 s, respectively (Figure 5E). Additionally, the frequent and large vibrational amplitudes were caused due to the poor vibration behavior of the HDPE composite. The results demonstrate that the addition of Si_3_N_4_ reduces and stabilizes the vibrational behavior of the HDPE composites, particularly with a 4 wt% addition of Si_3_N_4_. However, the excessive addition of Si_3_N_4_ resulted in severe vibrations, and this phenomenon might be attributed to the agglomeration of Si_3_N_4_ particles.

### 3.4. The Wear Rate Behavior of the Modified Composites

CLSM was used to observe the worn morphologies of the HDPE composite with different concentrations of Si_3_N_4_, and the results are presented in Figure 6. The wear scratch and its 3D profile results shown in Figure 6A, demonstrate that noticeable wear scratch occurred in the HDPE composites with 4 wt% Si_3_N_4_ after the friction test, and the worn scratch cross-section had a depth of 18 μm (Figure 6B). Figure 6C shows the different wear depths of the HDPE composites with different Si_3_N_4_ concentrations. The HDPE with only T-ZnOw presented a wear depth of 20.6 μm, while it marginally increased to 21.8 μm with the addition of 2 wt% Si_3_N_4_. However, the addition of 4 wt% Si_3_N_4_ decreased the wear depth to 18 μm, and the decreasing trend was observed with the additions of 6 wt% and 8 wt% Si_3_N_4_, which had wear depths of 17.1 μm and 15.6 μm, respectively. In addition, the wear volumes of the modified HDPE composites (Figure 6D) exhibited a slight increase from 0.0152 mm^3^ (0 wt% Si_3_N_4_ addition) to 0.0159 mm^3^ (2 wt% Si_3_N_4_ addition), and then significantly decreased with the increase in the Si_3_N_4_ concentration, which finally declined to the lowest value at 0.0128 mm^3^ with the addition of 8 wt% Si_3_N_4_. A similar trend was observed for the wear rate responses in Figure 6E. The wear rate of HDPE with only T-ZnOw was 3.3 × 10^−5^ mm^2^ *s/N, while it significantly declined to 3.0 × 10^−5^ mm^2^ *s/N with the addition of 6 wt% Si_3_N_4_ and 2.8 × 10^−5^ mm^2^ *s/N with the addition of 8 wt% Si_3_N_4_, after a slight increase with the addition of 2 wt% Si_3_N_4_. The wear rate results indicated that an increase in the Si_3_N_4_ content can enhance the wear-resistance of HDPE/T-ZnOw composites.

### 3.5. The Micro-Morphologies of the Modified Composites

It is known that the deformation on the polymer surface had a direct connection with the frictional status, such as the stick-slip deformations [37,38]. The micro-worn morphologies of the modified HDPE composites were observed using SEM, and the results are shown in Figure 7. It can be observed from Figure 7A that the deforming fringes were uniformly distributed on the worn surface and were perpendicular to the wear direction of the HDPE with only T-ZnOw. This exhibited stick-slip behavior and COF fluctuation during the friction process. However, this stick-slip behavior was weakened with the addition of 2 wt% Si_3_N_4_ to the HDPE, as observed in Figure 7B. Moreover, the worn surface was flat with a small waving deformation and grooves on the HDPE composite with 4 wt% Si_3_N_4_ (Figure 7C). The smooth and flat surface might lead to a stable COF process. However, significant grooves replaced the deforming fringes and became the dominant worn deformations on the worn surface with the addition of 6 wt% Si_3_N_4_ to the HDPE composite in Figure 7D. Additionally, when the Si_3_N_4_ content was increased to 8 wt%, noticeable cracks and grooves occurred on the HDPE worn surface, which increased the roughness and decreased the flatness (Figure 7E). The rough surface can result in a severe and unstable COF process. The micro-morphologies observed, demonstrated that the addition of 4 wt% Si_3_N_4_ can effectively reduce the deforming fringes on the worn surface of the HDPE composites. This can reduce the COF fluctuations and the frictional-induced vibration. However, an increase in the concentration of Si_3_N_4_ can change deforming fringes to severe grooves and cracks, which can result in fluctuations in the COF process.

## 4. Discussion

The results of COFs (Figure 4), vibrations (Figure 5), and micro-morphologies (Figure 8) demonstrate that the addition of 4 wt% of Si_3_N_4_ can significantly reduce the fluctuations of COF and the vibration of HDPE composites by reducing the surface deformation. The mechanism was analyzed and shown in Figure 8. A new peak of 1080 cm^−1^ on the infrared spectrometry (Figure 8A) and a new energy excitation peak in the XRD result (Figure 8B) verified the presence of a bond between Si and O on the worn surface of the HDPE composite, with 4 wt% Si_3_N_4_ during the friction and wear process in an aqueous environment [39,40]. The new chemical bonds meant that the origin of the Si–N chemical bond on the Si_3_N_4_ particle would be broken, and a new Si–O chemical bond would be created in an aqueous environment; the new Si–O bond would continue to react in an aqueous environment to form a complex compound that had lubricating effects (Figure 8C,D) [41,42,43]. Therefore, with the lubricating complex compound on the contacting surfaces for improving the lubricating condition, the surface deforming behaviors on the HDPE composite worn surface could be weakened (showed in Figure 7), and consequently reduced the vibration, COFs and its amplitudes (Figure 4 and Figure 5).

The micro-morphologies of the carbon steel balls are shown in Figure 9A–C. Figure 9A shows a slight scratch on the carbon steel ball after conducting the friction test on the HDPE composite with 4 wt% Si_3_N_4_, and it exhibited a smooth wear process. However, a few visible cracks and scratches occurred on the carbon steel ball after the friction test was conducted on the HDPE composite with 6 wt% Si_3_N_4_, which resulted in a rough surface (Figure 9B). Furthermore, the carbon steel ball worn against the HDPE composite with 8 wt% Si_3_N_4_ became significantly rough with severe cracks and grooves (Figure 9C), and the rough surface resulted in a fluctuating and unstable COF. Additionally, the vibration responses and amplitudes were unstable. Figure 9D,E present the element distribution maps of the worn surfaces of the HDPE composites with 4 wt% and 8 wt% Si_3_N_4_, respectively. The uniform distribution of Si and Zn on the worn surface of the HDPE composite after a 4 wt% Si_3_N_4_ addition (Figure 9D), contributed to more than 25% reductions in the COF and vibration, as well as more than 70% reductions in their amplitudes individually (Figure 4 and Figure 5). However, the significant aggregation of Si on the severely cracked and torn areas proved that the aggregating phenomenon of Si_3_N_4_ particles in high concentrations (such as 6 wt%, 8 wt%) led to severe deformation, which was in contrast to the occurrence of deforming fringes on the worn surface (Figure 9E). Moreover, the aggregating phenomenon observed in the composites with high concentration of Si_3_N_4_, led to severe surface deformation on the HDPE composites, and reduced its yield stress and strain hardening properties (Figure 3). The wear-resistance due to higher yield stress and strain hardening [44,45], reduced the surface deformation in combination with the lubricating effect due to a 4 wt% addition of Si_3_N_4_. However, the reduced strain–stress response and aggregating phenomenon of Si_3_N_4_ at high concentrations resulted in a rough and worn surface with severe cracking and tearing deformations. Therefore, this resulted in a sharp increase in COF, which reached 0.11 with severe fluctuations from 0.1 to 0.12, and unstable vibrational behaviors with significant vibration points (Figure 4 and Figure 5).

## 5. Conclusions

HDPE composites with T-ZnOw as a fixed enhancer and Si_3_N_4_ particles were developed to reveal the frictional responses in an aqueous environment. The complex compound reacted from the Si_3_N_4_ that existed between contact surfaces and functioned as a lubricating film to enhance lubrication. Furthermore, the higher yield stress and strain hardening responses in a 4 wt% Si_3_N_4_ addition improved the wear-resistance; therefore, the decrease in surface deformation led to more than 25% in COF and vibration individually, and also a maximum of 75% reductions in the COF amplitudes (ΔA). However, the aggregation phenomenon and the reduced strain–stress responses at a high concentration of Si_3_N_4_ disrupted the positive reduction effects and resulted in a rough and worn surface with severe cracking and tearing deformations. Therefore, this resulted in exacerbating COF and vibration behaviors, which sharply increased to 0.11 in COF and −1.3–1.2 m/s^2^ in the vibration amplitudes. The results gained herein can provide theoretical support for adopting Si_3_N_4_ particles as a new lubricating filler in an aqueous environment, and reveal the synergistic effect of two fillers on improving the tribological property of polymers.

## Figures and Tables

**Figure 1 polymers-14-00433-f001:**
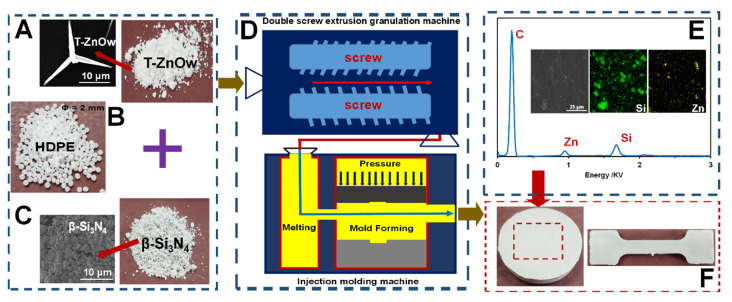
Experimental materials and fabrication: (**A**) tetra-needle-like ZnO particles (T-ZnOw); (**B**) high-density polyethylene particle (HDPE); (**C**) Si_3_N_4_ nanoparticle; (**D**) double screw extrusion granulation and injection molding machines; (**E**) the EDS curve and element distribution maps of HDPE composite with 6 wt% Si_3_N_4_ addition; and (**F**) the samples fabricated for friction and mechanical tests.

**Figure 2 polymers-14-00433-f002:**
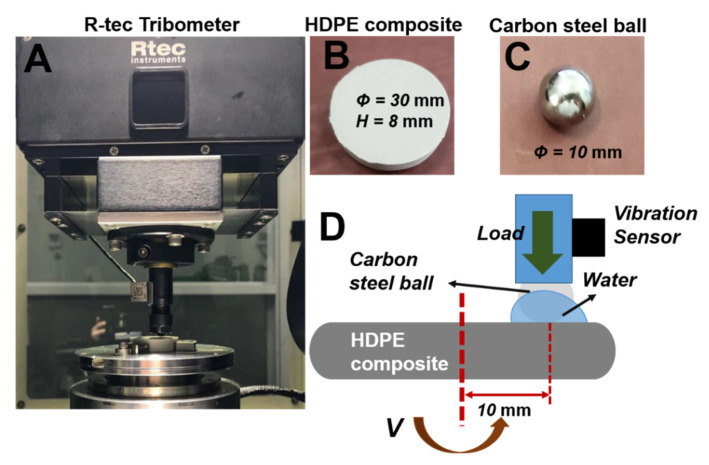
Testing apparatus and method: (**A**) R–tec tribo-meter and rotation friction model; (**B**) the modified HDPE composite; (**C**) the 10 mm carbon steel frictional pair ball; and (**D**) schematic diagram of wear and vibration experiments in aqueous environment.

**Figure 3 polymers-14-00433-f003:**
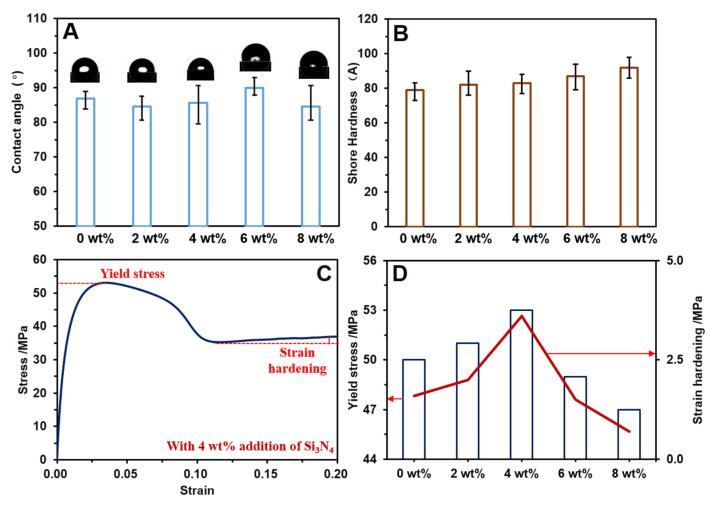
Properties of the modified HDPE composites: (**A**) the contact angle with distilled water; (**B**) the shore hardness; (**C**) the strain–stress curve of the HDPE composite with 8 wt% Si_3_N_4_ addition; and (**D**) the yield stress and strain hardening of modified HDPE composites.

**Figure 4 polymers-14-00433-f004:**
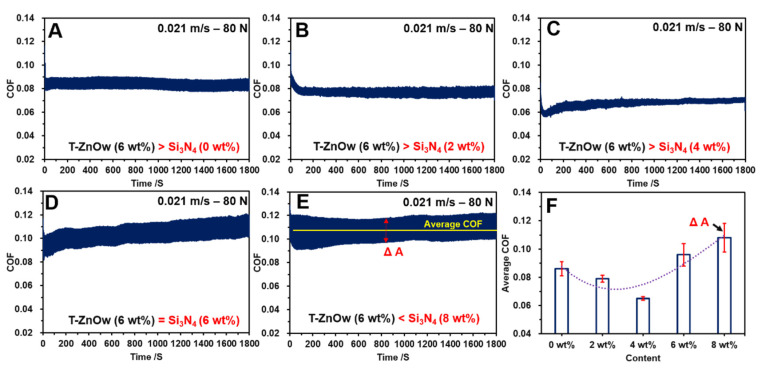
The COF results and the amplitudes of the modified HDPE composites with different Si_3_N_4_ concentrations: (**A**) 0 wt% Si_3_N_4_ addition; (**B**) 2 wt% Si_3_N_4_ addition; (**C**) 4 wt% Si_3_N_4_ addition; (**D**) 6 wt% Si_3_N_4_ addition; (**E**) 8 wt% Si_3_N_4_ addition; and (**F**) the average COF results and its amplitudes.

**Figure 5 polymers-14-00433-f005:**
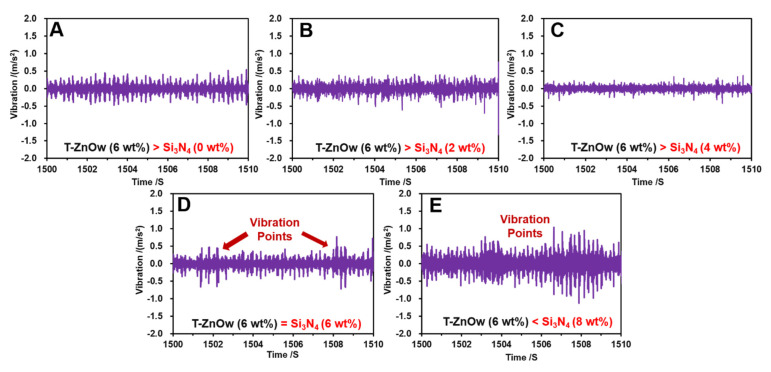
The friction-induced vibration responses of the modified HDPE composites with different contents of Si_3_N_4_ at 0.021 m/s and 80 N in a water-lubricated environment: (**A**) 0 wt% Si_3_N_4_ addition; (**B**) 2 wt% Si_3_N_4_ addition; (**C**) 4 wt% Si_3_N_4_ addition; (**D**) 6 wt% Si_3_N_4_ addition; and (**E**) 8 wt% Si_3_N_4_ addition.

**Figure 6 polymers-14-00433-f006:**
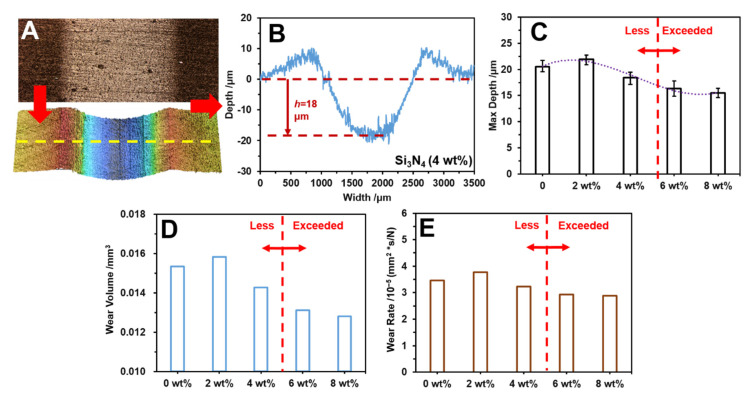
The wear rate behavior of HDPE composites with different concentrations of Si_3_N_4_ at 0.021 m/s and 80 N in an aqueous environment: (**A**) wear scratch and its 3D profile; (**B**) cross profiles of the wear scratch with 4 wt% Si_3_N_4_ addition in the composite; (**C**) the maximum depths of the modified HDPE composites; (**D**) the wear volumes of the modified HDPE composites; and (**E**) the wear rates of the modified HDPE composites.

**Figure 7 polymers-14-00433-f007:**
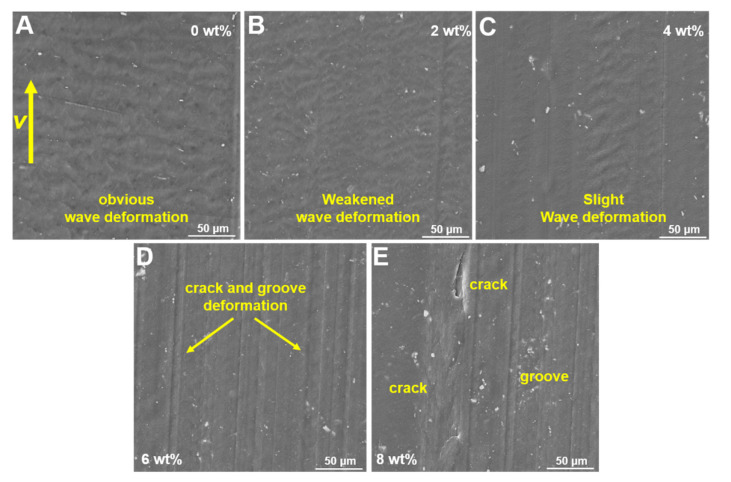
The micro-morphologies of the HDPE composites with different Si_3_N_4_ contents: (**A**) 0 wt% Si_3_N_4_ addition; (**B**) 2 wt% Si_3_N_4_ addition; (**C**) 4 wt% Si_3_N_4_ addition; (**D**) 6 wt% Si_3_N_4_ addition; and (**E**) 8 wt% Si_3_N_4_ addition.

**Figure 8 polymers-14-00433-f008:**
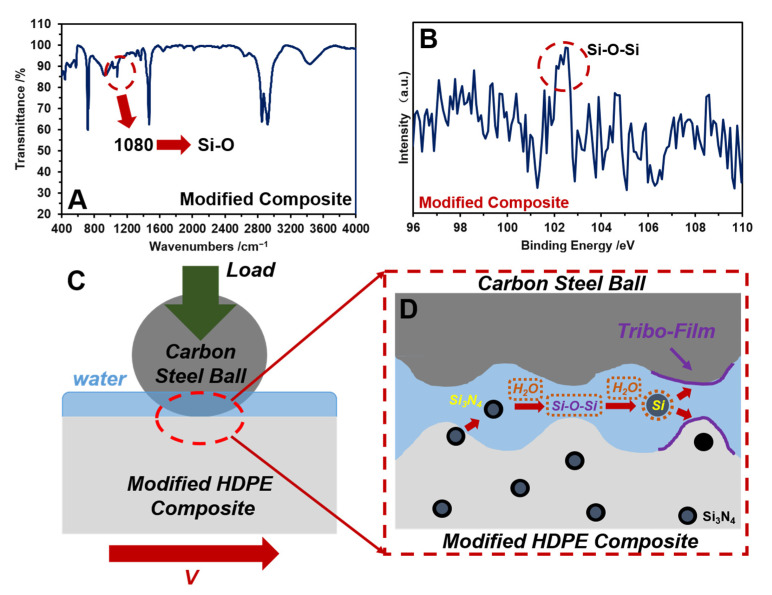
Mechanism of Si_3_N_4_ for the enhancement of lubrication: (**A**) infrared spectrometry; (**B**) diffraction of X-rays results of the worn surface with 4 wt% Si_3_N_4_ addition in the HDPE composite (**C**); and (**D**) are the lubricating mechanism of 4 wt% Si_3_N_4_ addition in the HDPE composite during friction.

**Figure 9 polymers-14-00433-f009:**
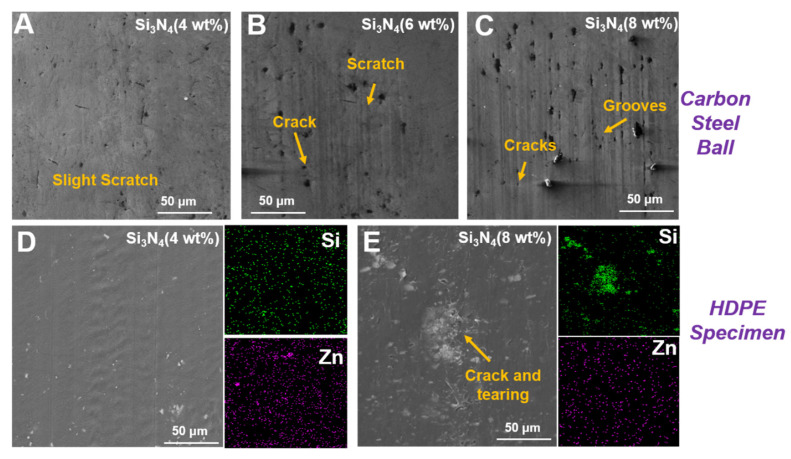
Micro-morphologies of the carbon steel ball with different modified HDPE composites: (**A**) 4 wt% Si_3_N_4_ addition; (**B**) 6 wt% Si_3_N_4_ addition; and (**C**) 8 wt% Si_3_N_4_ addition. The element distributions on the polymer plates with different Si_3_N_4_ concentrations: (**D**) 4 wt% addition and (**E**) 8 wt% addition.

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
