# Peer review of "The Reduction in the Deformation of HDPE Composites Using Self-Lubricating Fillers in an Aqueous Environment"

_polymers, 2022, doi:10.3390/polym14030433_

Round 1

Reviewer 1 Report

Dear Editor,

The topic of the paper is interesting and suits the Journal of MDPI Polymers. However, a major revision is required before this manuscript is qualified to be published in this prestigious journal. The manuscript is needed to be revised grammatically. The authors are required to check the whole manuscript with a grammar specialist as it has several grammatical errors. Only after revising the manuscript based on the comments, the paper is suggested to be published in MDPI. Further information on various issues identified in the manuscript appears below:

  1. The introduction section needs to be revised. A paragraph should be dedicated to the importance of your work.
  2. The authors have done a great job on the literature review. However, the introduction needs more attention. More information on new materials related to the topic of this paper can be found here:

"Fracture Properties Evaluation of Cellulose Nanocrystals Cement Paste." Materials 13, no. 11 (2020): 2507.

  1. Please provide more detailed reasoning behind the behavior. The details should include rigid numbers or percentages.
  2. Please indicate how many samples for each experiment have been used. Please revise the other experiments respectively.
  3. Please describe the process of each experiment. Also indicate the model of each tool that is used in the experiment. What is the accuracy of each machine? Please explain them accurately.
  4. The conclusion needs more elaboration. Please use more sentences containing percentages and illustrate the main conclusions in the manuscript. Please paraphrase your results and discussions and use them in the conclusion part.

Author Response

Response to Reviewer 1 Comments

Point 1: The introduction section needs to be revised. A paragraph should be dedicated to the importance of your work.

Response 1: Thanks for the reviewer’s good comment on improving the quality of the introduction section. The author had ultimately improved the introduction part for highlighting the innovation and importance of this work. A paragraph for dedicating the importance of our work had been enhanced and is shown below:

“Although the lubricating fillers and flower-like fillers have been used to modify materials to reduce surface deformation individually, the synergistic effect of these two different fillers on improving the tribological property has been rarely conducted. Therefore, in this study, the T-ZnOw particles, which proved excellent deforming resistance on the polymer surface, were adopted as an enhancer with static content on the HDPE composites. And the Si3N4 particle was selected as a lubricating enhancer due to its stable physical properties and exceptional chemical reacting potential in the aqueous environment [33, 34]. The results obtained in this study can explore the possibility of Si3N4 as a new lubricating filler in an aqueous environment and reveal the synergistic effect of two different fillers on improving tribological property in the polymeric material.”

Point 2: The authors have done a great job on the literature review. However, the introduction needs more attention. More information on new materials related to the topic of this paper can be found here:

"Fracture Properties Evaluation of Cellulose Nanocrystals Cement Paste." Materials 13, no. 11 (2020): 2507.

Response 2: Thanks for the reviewer’s comment. The author had done the literature review again, and some novel and related literature had been cited in this paper. In addition, the author had improved the introduction part to emphasize the novelty and importance of our work.

Point 3: Please provide more detailed reasoning behind the behavior. The details should include rigid numbers or percentages.

Response 3: Thanks for the reviewer’s professional comment, it was the author’s fault that we had not provided detailed reasoning behind the behavior. In our study, the Si3N4 was adopted to modify the HDPE-T-ZnOw composites to explore its possibility of working as a new lubricating filler in an aqueous environment. The results showed that the addition of Si3N4 had reduced the COF, vibration, and their amplitudes notably, and the excellent performance happened on 4 wt% Si3N4 addition. The XRD and infrared spectrometry results proved a new complex compound, reacted by Si3N4, happened on the contacting surfaces and worked as tribol-film. In addition, the highest yield stress and strain hardening, contributed by 4 wt% Si3N4 addition, led to outstanding wear resistance on HDPE composite. Therefore, with the tribol-film and good wear resistance, the HDPE with 4 wt% Si3N4 addition showed excellent performance on the tribological property with the lowest and the most stable COF and vibration. However, the increasing addition of Si3N4 (like 6 wt% and 8 wt%) caused the aggregation phenomenon, which weakened the wear resistance and led to severe deformation on the worn surface, thus breaking the lubricating effect and exacerbating the friction process with high and fluctuating COF and vibration behaviors.

The author had ultimately improved this part in this manuscript for explaining the behaviors with more evidence, and the academic expression had been enhanced.

Point 4: Please indicate how many samples for each experiment have been used. Please revise the other experiments respectively.

Response 4: Thanks for the reviewer’s comment, it was the author’s fault that we had not put enough emphasis on how many samples were used in each experiment. We tested three samples in each experiment to ensure repeatability and averaged all data for the results. The author had emphasized this part in section 2.2, which showed below:

“A new polymer plate and carbon steel ball were used for each experiment. In addition, all tests in this study were repeated thrice to ensure repeatability.”

Point 5: Please describe the process of each experiment. Also, indicate the model of each tool used in the experiment. What is the accuracy of each machine? Please explain them accurately.

Response 5: Thanks for the reviewer’s comment, it was the author’s fault that we had not explained the detailed information of the experiment. We have two main experiments in this study: frictional and vibrational experiments.

For the frictional experiment, a commercial ball-on-pin friction testing machine (R-tec Tribo-meter, Rtec Instruments Inc, USA) was used. During friction test, a 10-mm diameter carbon steel ball was maintained stationary at the top, while the lower HDPE specimens were rotated at a velocity of 0.021 m/s, and the vertical load was 80 N, and the radius of rotation was set to 10 mm (showed in Fig. 2). The experiment lasted 1800 s, and the test data were recorded at an interval of 0.001 s. The frictional coefficient (COF) result was obtained directly.

The vibrational senor from Bruel & Kjaer testing machine (Sensor 4535-B, B & K Ltd, Denmark) was used to collect the vibration data for the vibrational experiment. The vibrational sensor was fixed on the upper fixture during the friction process to measure the vibration induced in the HDPE-based composites during the wear and friction process (shown in Fig. 2 (D)). The experiment lasted 1800 s, and the test data were recorded at 1000 Hz (an interval of 0.001 s).

Other experiments, like the tensile test, morphology observation of worn surface, and chemical reaction experiment, were completed under the operating requirements.

The author had accurately explained the experimental apparatus (section 2.2), measurement techniques, and procedures (section 2.3) in this revision.

Point 6: The conclusion needs more elaboration. Please use more sentences containing percentages and illustrate the main conclusions in the manuscript. Please paraphrase your results and discussions and use them in the conclusion part.

Response 6: Thanks for the reviewer’s comment. The author had improved the conclusion part, which is shown below:

“HDPE composites with T-ZnOw as a fixed enhancer and Si3N4 particles were developed to reveal the frictional responses in an aqueous environment. The complex compound reacted from the Si3N4 between contact surfaces and functioned as a lubricating film to enhance lubrication. Besides, the higher yield stress and strain hardening responses in 4 wt% Si3N4 addition improved the wear resistance. Therefore, the decrease in surface deformation led to more than 25 % in COF and vibration individually, and also a maximum of 75 %reductions on the COF amplitudes (â–³A). However, the aggregation phenomenon and the reduced strain-stress responses at a high concentration of Si3N4 disrupted the positive reduction effects and resulted in a rough and worn surface with severe cracking and tearing deformations. Therefore, this exacerbated COF and vibration behaviors which sharply increased to 0.11 in COF and -1.3–1.2 m/s2 in vibration amplitudes. The results gained herein can provide theoretical support for adopting Si3N4 particles as a new lubricating filler in an aqueous environment and reveal the synergistic effect of two fillers on improving tribological property on polymers.”

Reviewer 2 Report

I have reviewed the manuscript entitled "Reduction of deformation of HDPE composites using self-lubrication fillers in an aqueous environment" by Chuanbo Lin, et al., which was submitted to Polymers.  This manuscript shows a new filler used in HDPE composites for lubrication in an aqueous conditions.

Si3N4 ws used as a filler, and was composited with HDPE in concentrations of 0, 4, 6, and 8 wt%.  The composites were analyzed the friction properties using ball-on-pin friction testing machine, the surface topography using confocal laser scanning microscopy, and EDX.  Surface hydrophilicity and surface roughness of the composites also were scanned using contact angl e meter (DSA 100) and a white light interferometer (Micro Xam), respectively.  After chemical and surface characterization of the HDPE composites with Si3N4 particles, mechanical properties such as coefficient of friction, friction-induced vibration response, and wear rate behavior of the HDPE composites in aqueous environment were investigated and discussed on a possibility of the polymer composited with low friction-induce fluctuation.

This paper is well organized and also discussed well, and provides new useful information on a possibility of polymer composited with low friction-induce fluctuation.  From the reviewing results, I evaluate that the paper is acceptable for publication.

Author Response

Thanks so much for your very positive comments on our work. 
